# Investigations of Abrasive Wear Behaviour of Hybrid High-Boron Multi-Component Alloys: Effect of Boron and Carbon Contents by the Factorial Design Method

**DOI:** 10.3390/ma16062530

**Published:** 2023-03-22

**Authors:** Yuliia Chabak, Ivan Petryshynets, Vasily Efremenko, Michail Golinskyi, Kazumichi Shimizu, Vadym Zurnadzhy, Ivan Sili, Hossam Halfa, Bohdan Efremenko, Viktor Puchy

**Affiliations:** 1Physics Department, Pryazovskyi State Technical University, 87555 Mariupol, Ukraine; 2Institute of Materials Research, Slovak Academy of Sciences, 04001 Kosice, Slovakia; 3Muroran Institute of Technology, Muroran 050-8585, Hokkaido, Japan; 4Central Metallurgical Research and Development Institute, Eltebbin, Helwan 12422, Cairo, Egypt

**Keywords:** multi-component alloy, carbon, boron, wear resistance, hardness

## Abstract

This paper is devoted to the evaluation of the “three-body-abrasion” wear behaviour of (wt.%) 5W–5Mo–5V–10Cr-2.5Ti-Fe (balance) multi-component (C + B)-added alloys in the as-cast condition. The carbon (0.3 wt.%, 0.7 wt.%, 1.1 wt.%) and boron (1.5 wt.%, 2.5 wt.%, 3.5 wt.%) contents were selected using a full factorial (3^2^) design method. The alloys had a near-eutectic (at 1.5 wt.% B) or hyper-eutectic (at 2.5–3.5 wt.% B) structure. The structural micro-constituents were (in different combinations): (a) (W, Mo, and V)-rich borocarbide M_2_(B,C)_5_ as the coarse primary prismatoids or as the fibres of a “Chinese-script” eutectic, (b) Ti-rich carboboride M(C,B) with a dispersed equiaxed shape, (c) Cr-rich carboboride M_7_(C,B)_3_ as the plates of a “rosette”-like eutectic, and (d) Fe-rich boroncementite (M_3_(C,B)) as the plates of “coarse-net” and ledeburite eutectics. The metallic matrix was ferrite (at 0.3–1.1 wt.% C and 1.5 wt.% B) and “ferrite + pearlite” or martensite (at 0.7–1.1 wt.% C and 2.5–3.5 wt.% B). The bulk hardness varied from 29 HRC (0.3 wt.% C–1.5 wt.% B) to 53.5 HRC (1.1 wt.% C–3.5 wt.% B). The wear test results were mathematically processed and the regression equation of the wear rate as a function of the carbon and boron contents was derived and analysed. At any carbon content, the lowest wear rate was attributed to the alloy with 1.5 wt.% B. Adding 2.5 wt.% B led to an increase in the wear rate because of the appearance of coarse primary borocarbides (M_2_(B,C)_5_), which were prone to chipping and spalling-off under abrasion. At a higher boron content (3.5 wt.%), the wear rate decreased due to the increase in the volume fraction of the eutectic carboborides. The optimal chemical composition was found to be 1.1 wt.% C–1.5 wt.% B with a near-eutectic structure with about 35 vol.% of hard inclusions (M_2_(B,C)_5_, M(C,B), M_3_(C,B), and M_7_(C,B)_3_) in total. The effect of carbon and boron on the abrasive behaviour of the multi-component cast alloys with respect to the alloys’ structure is discussed, and the mechanism of wear for these alloys is proposed.

## 1. Introduction

The problem of the wear of machine parts is becoming more and more serious in view of the intensification of the technological processes in different areas that deal with the processing of raw materials. This leads to increasing economic losses and energy consumption, with a corresponding increase in greenhouse gas emissions [1]. Consequently, different approaches are utilised to reduce wear losses, including the development of functional alloys with improved tribological properties [2,3]. Using advanced cast alloys remains one of the most important approaches to prolonging the lifetime of machine parts subject to severe abrasive and erosive wear under operating conditions. Among these alloys, the high-alloyed cast irons stand out for their exceptional wear resistance and fair processability [4,5,6]. They include a group of multi-component cast irons (MCCIs) developed by Matsubara et al. in the mid-1990s [7,8]. The idea of MCCIs is based on the concept of AISI M2 (T11302)-grade high-speed steel (adding W, Mo, V, and Cr in approximately equal proportions (2–5 wt.%)) with a doubled carbon content, which allowed the development of wear-resistant materials for the rolls of hot rolling mills. After an appropriate heat treatment, the MCCIs demonstrate a composite multi-phase structure consisting of hard compounds (MC, M_7_C_3_, M_2_C, M_6_C) dispersed in the martensite matrix [9]. The improved abrasive resistance of MCCIs is highlighted by Inthidech et al. [10,11] and Opapaiboon et al. [12]. The mechanism of wear of MCCIs under abrasion by soft abrasive particles is presented by de Mello and Polycarpou [13]. The research group of Shimizu systematically studied the abrasive and high-temperature erosion behaviours of MCCIs [14,15,16,17]. They emphasised the high potential of MCCIs under erosion by hard particles (alumina) at temperatures up to 900 °C. The data from the laboratory testing of MCCIs are confirmed by the successful usage of MCCIs in metallurgy and the cement industry [18].

Another up-to-date approach in abrasive-resistant alloys involves “high-boron cast irons” (HBCIs); this concept was originally proposed by Lakeland [19]. The idea of HBCIs was to replace the cementite carbide (Fe_3_C) in wear-resistant Fe-C alloys with a harder boride (Fe_2_B) [20]. For this purpose, in the Fe-C alloy, carbon was partially replaced by boron: HBCIs initially contained 1.2–3.5 wt.% B and 0.2–0.5 wt.% C to create an Fe_2_B-based eutectic (plain Fe-B alloys) [21,22]. The superiority of Fe_2_B compared to Fe_3_C in terms of hardness is due to the stronger hybridisation of the “metal–boron” bond compared with the “metal–carbon” bond [23]. The microstructure and abrasive wear resistance of the plain Fe-B alloys were studied in [24,25,26,27,28]. These works describe the HBCIs’ structure as a continuous skeleton consisting of an Fe_2_B-based eutectic surrounding the dendrites of the matrix phase (ferrite, pearlite, or martensite). Zhang et al. [29] deduced that under “two-body-abrasion’ conditions, Fe-C-B high-boron alloys with 0.3–1.0 wt.% C and 1.5–3.0 wt.% B significantly overperform high-chromium cast iron in terms of the wear behaviour. Fu et al. [30,31] reported that under “two-body-abrasion” testing, the Fe-B-C alloys (0.08–0.2 wt.% C; 2.2–4.0 wt.% B) have an advantage over the bearing steel GCr15 and Ni-hard cast iron [30], while hardened Fe-C-B alloys (0.1–0.25 wt.% C; 0.8–1.2 wt.% B; 56 HRC) showed a similar wear response to the high-chromium cast iron (15 wt.% Cr; 62.5 HRC) [31]. Hot forging leads to broken boride networks, thus increasing HBCIs’ impact toughness [32] and “three-body-abrasion” wear resistance as compared to cast unbroken boride skeletons [33]. The successful application of high-boron alloys in the form of various parts (liners, hummers, guides, etc.) in actual “in-field” conditions is reported by Ma and Zhang [34]. Researchers have attempted to improve the tribological properties of plain Fe-B alloys by adding alloying elements (Cr, Mn, Mo, Ti, etc.); Cr is most often used [35,36,37,38,39,40]. Jian et al. [35,36] found that adding 2 wt.% Cr increases the wear resistance of an Fe-3.0 wt.% B alloy under SiC and SiO_2_ abrasion; if the Cr content is increased further, the wear resistance decreases. Similarly, the “two-body-abrasion” wear behaviour of Fe-B alloys after Cr was added was reported in [37,38]; the positive effect of Cr is explained by the improvement of the boride Fe_2_B fracture toughness [38,39,40,41,42]. Jian et al. [43,44] studied the effect of manganese and deduced that adding 2.0 wt.% Mn into an Fe-B-C alloy increases its “three-body-abrasive” wear resistance under different applied loads, mostly because it enhances the fracture toughness of Fe_2_B particles. The effect of boron on the microstructure and properties of high-Cr cast irons was described in many works [45,46,47,48]; the increased hardness and wear resistance of the B-added alloys were highlighted.

The authors of the present paper recently proposed the novel (“hybrid”) concept of a wear-resistant alloy that combines the “multi-component” and “high-boron” approaches [49]. It involves multi-alloying by several strong carbide-forming elements (W, Mo, V, Cr, and Ti) and the partial replacement of carbon with an increased amount of boron. This concept differs from the high-B multi-component Fe-B-C-Cr-Mo-Al-Si-V-Mn-Ti alloys studied in [49,50] in that it involves a higher carbon content and different key elements (W, V, Cr, and Ti). The structure characterisation and thermodynamic analysis of “hybrid” MCCIs with variable carbon (0.3–1.1 wt.%) and boron (1.5–3.5 wt.%) contents are described in detail in [51,52,53]. These alloys are intended for wear applications, and therefore a comprehensive tribological assessment of these alloys under different wear conditions is strongly desired. This work is aimed at the evaluation of the ‘three-body-abrasion’ behaviour of as-cast high-boron Fe-C-W-Mo-V-Cr-Ti alloys, focusing on the “structure/wear resistance” correlation with respect to the carbon and boron contents.

## 2. Materials and Methods

In the present study, nine “hybrid” high-boron multi-component alloys were investigated. All alloys were intended to have the same basic contents of alloying elements (5 wt.% W, 5 wt.% Mo, 5 wt.% V, 10 wt.% Cr, 2.5 wt.% Ti, 1 wt.% Mn, 1 wt.% Si) and different C and B concentrations. These concentrations were set in accordance with the three-level full factorial design of experiments 3^2^, meaning nine experimental alloys [54,55,56]. Two variables—the carbon content (factor F_1_) and the boron content (factor F_2_)—were varied on three equidistant levels, called the “lower level”, “middle level”, and “upper level” (Table 1). The nominal values of the C content were 0.3 wt.%, 0.7 wt.%, and 1.1 wt.%. At each of these levels, the B content was 1.5 wt.%, 2.5 wt.%, and 3.5 wt.%. The selected nominal and coded values of the variables [55] are presented in Table 1; the matrix of the experiment is shown in Table 2.

The alloys were prepared by air-melting 50 kg of raw material (master alloys, pig iron, and steel scrap) in a high-frequency induction furnace. The melt was poured into a Y-block sand mould. The chemical compositions of the alloys were detected using a SPECTROLAB analyser (AMETEK, Inc., Berwyn, PA, USA); the results are shown in Table 2. Specimens with a size of 6 × 12 × 25 mm were cut from the ingots using high-speed precision cutting equipment (Refinetech Co., Ltd., RCA-234, Kanagawa, Japan). The alloys were studied in the as-cast condition.

The specimens for the microstructural study were mirror-polished and etched by a 5 vol.% nital etchant. After the specimens were polished with SiC emery papers, they were gently polished with Al_2_O_3_-containing suspensions to prevent the breakage of boride phases. The microstructure was characterised using the optical microscopes (OMs) “Eclipse M200” (Nikon, Tokyo, Japan) and “GX71” (Olympus, Tokyo, Japan), as well as the scanning electron microscopes (SEMs) “JSM-7000F” and “JSM-6510” (both JEOL, Tokyo, Japan). The phase chemical composition was studied through energy-dispersive X-ray spectroscopy (EDS) using the detector ‘JED-2300’ (JEOL, Tokyo, Japan). The area fraction of the structural components was calculated by applying the Rosiwal lineal method to ten optical micrographs with dimensions of 440 × 600 μm, with further averaging of the results. The bulk hardness of the alloys was measured according to the Rockwell method (scale C) using the FR-X1 tester (Future-Tech, Kanagawa, Japan). Five hardness measurements were performed on each of three different specimens (a total of 15 measurements for each alloy), with further averaging of the results. The data from the XRD/EDS study of the phase constituents of the alloys were adopted from [51,52].

The wear resistance of the alloys was evaluated using the “three-body-abrasion” tester built at Pryazovskyi State Technical University (Mariupol, Ukraine). Before abrasive wear testing, the surface of the specimens was ground with sandpaper to an R_a_ value of ~0.6 μm. During the abrasion test, the specimen was pressed by a 40 mm diameter rubber roller with a force of 20 N. The roller rotated at a speed of 10.8 s^−1^. Alumina particles with diameters of about 0.5 mm were fed into the gap between the roller and the specimen’s surface at a feed rate of 0.75 kg·min^−1^. The total test duration was 1800 s. After the test, the specimen was alcohol-cleaned and the mass loss (Δ*m* (g)) was measured by weighing the specimen on an electronic balance with an accuracy of 0.0001 g. Each test was repeated three times and the results were averaged. The wear resistance was estimated using the wear rate (*WR*), which was calculated as follows:(1)$$WR=\frac{\Delta m}{b\cdot t},$$
where *b* is the specimen width (mm) and *t* is the test duration (s).

## 3. Results and Discussion

### 3.1. Microstructure Characterisation

The microstructures of the alloys are shown in Figure 1 and Figure 2. The images in Figure 1 are shown in a back-scattered electron (BSE) mode to more clearly distinguish the various phase constituents that differ in terms of the elemental composition (i.e., the BSE contrast). At the minimum carbon (0.3 wt.%) and boron (1.5 wt.%) contents, the structure shows near-eutectic patterns consisting of the eutectic colonies (as shown in Figure 1a), with no signs of matrix γ-phase dendrites or primary boride inclusions. The eutectic colonies are bunches of fibres branched in all directions within the metallic matrix (ferrite). The fibres were characterised in earlier works [51,52] as borocarbide M_2_(B,C)_5_ containing W (12–30 wt.%), Mo (14–19 wt.%), V (15–18 wt.%), Cr (10–21 wt.%), and Ti (1–5 wt.%) (the exact content of each element depends on C and B concentrations in alloy). Due to enrichment with elements of a higher Z number (W and Mo), the fibres have a light BSE contrast that differs from the dark BSE contrast of a matrix enriched with the elements of a lower Z number (Fe, Cr, Mn, and Si). The fibres radiate from the colony’s centre at different angles, often bending at a right angle with the formation of “square tubes” (shown in Figure 2a). The specific branching and bending of the fibres create a distinctive eutectic pattern called a “Chinese-script” pattern [57,58]. The growth of the colonies during solidification was limited by other colonies. In the places where they collided, a kind of matrix “rim” appeared that had a less-dense arrangement of eutectic fibres (Figure 1a).

At 0.3 wt.% C, increasing the boron content to 2.5 wt.% changed the structural status to hyper-eutectic, as shown by the emergence of coarse (tens of microns in size) primary inclusions of borocarbide M_2_(B,C)_5_ with a prismatic shape (Figure 1b). Many of these inclusions have an axial hole filled with the matrix phase (Figure 2b). Their chemical composition is close to that of the aforementioned eutectic fibres [51,52]. The space between the prismatoids is still filled with the “Chinese-script” eutectic colonies, although they became very sparse when the amount of fibres decreases (the width of the matrix inter-layers increases accordingly).

An increase in the boron content to a maximum (3.5 wt.%) resulted in an increase in the number of primary borocarbides. Furthermore, eutectics of another morphological type that was not the “Chinese-script” type appeared. This type has a “rosette”-like pattern characteristic of Cr_7_C_3_-based eutectics in high-Cr cast irons [48] (Figure 1c). The lamellae of the “rosette”-like eutectic differ from the primary inclusions due to their darker BSE contrast, indicating their enrichment in chromium (~31–45 wt.%) and iron (~30–40 wt.%) [51,52]. The colonies of the “rosette” eutectic are the plates (carboboride M_7_(C,B)_3_ [51]) that radiate in tens microns through the matrix from the colony’s centre (shown in the longitudinal direction on the right side of Figure 2c). These plates have a 3D-branched skeleton with a “rosette”-like pattern in cross-section (seen as the “petals” diverging from the centre, which are shown on the left side of Figure 2c).

In Figure 1c, the fine equiaxed particles can be clearly seen in the structure of the 0.3C–3.5B alloy thanks to their dark BSE contrast due to their enrichment in titanium. As shown in [51,52], they are Ti-based carboboride (M(C,B)) particles that were the first to crystallise from the melt due to the addition of 2.5 wt.% Ti. The titanium content varies in M(C,B) in a rather wide range (54–72 wt.% [51,52]), depending on the enrichment of the inclusion’s core with titanium (revealed in the BSE image as a dark-contrast inner area; Figure 2d). These particles are present in all alloys; they are spread within the matrix areas and eutectic colonies and inside the primary carboborides, although mostly they are concentrated along the primary borocarbide contour. M(C,B) particles feature an angular cuboid shape characteristic of compounds with a cubic (NaCl-type) lattice.

The alloy 0.7C–1.5B showed a near-eutectic structure similar to that of 0.3C–1.5B with one distinction: two eutectics, the “Chinese-script” and “rosette” eutectics (with the majority of the former), were present (Figure 1d). Adding 2.5–3.5 wt.% B into the 0.7 wt.% C-alloy changed its status to hyper-eutectic, as confirmed by the formation of the primary M_2_(B,C)_5_ prismatoids (shown in Figure 1e,f). A minor amount of the “rosette” eutectic was observed in the 0.7C–2.5B alloy, along with the “Chinese-script” eutectic (major). In the 0.7C–3.5B alloy, the “Chinese-script” eutectic was absent while the morphology of (Cr/Fe)-rich eutectic partially transformed from “rosette” into a “fish-bone” (Figure 1f) or a coarse network with the lowered chromium content (20–22 wt.%) [51] in the eutectic plates.

The above-described effect of boron on the structure of alloys, in general, was also noted for the alloys with a maximum carbon content (1.1 wt.%), although they did have some distinctive features. The 1.1C–1.5B alloy retained a near-eutectic status, similar to other alloys with 1.5 wt.% B, while a “rosette” eutectic was mostly replaced by a ledeburite-like eutectic (as compared to the 0.7C–1.5B alloy). The features of the latter are shown in Figure 2e to reveal a classic “honeycomb” pattern [59], representing the portions of the Fe phase (ferrite and austenite) encapsulated (enclosed) within the hard matrix (borocementite M_3_(B,C)). Such a mutual arrangement of soft and hard phases makes it possible to clearly differentiate the ledeburite eutectic from the “Chinese-script” or “rosette” eutectics where hard fibres (lamellas) are branched within a softer matrix phase. The hard phase of the ledeburite eutectic is enriched by Fe (55–60 wt.%) and Cr (20–25 wt.%) (it also contains 4–5 wt.% of each of W, Mo, and V) [51]. In the hyper-eutectic 1.1C–2.5B and 1.1C–3.5B alloys, the ledeburite eutectic was replaced by a so-called “coarse-net” eutectic that shows coarse plates of borocementite M_3_(C,B) interconnected in the continuous network (skeleton) surrounding the matrix grains (Figure 2f). The lamellae of the “coarse-net” eutectic are heterogeneous in terms of their Cr and Fe contents, which vary between 12–22 wt.% and 67–75 wt.%, respectively [51]. The representative EDS spectra obtained from the aforementioned hard compounds are shown in Figure 3, revealing significant differences in their chemical compositions.

Table 3 contains the data on each alloy’s structure. The data include (a) the sets of structural constituents, (b) their volume fractions, and (c) the total volume fractions (T) of hard particles. (The latter were calculated by excluding the volume fraction of the matrix layers/areas belonging to the eutectic colonies.) Table 3 shows that boron promotes the growth of the total volume fraction of the hard inclusion, and this effect is stronger for higher carbon contents. As follows from Table 3, all near-eutectic alloys (with 1.5 wt.% B) and the 0.3C–2.5B alloy contained ferrite as a metallic matrix. At higher carbon contents, the alloys with 2.5 wt.% B had a martensite matrix and the alloys with 3.5 wt.% B had a “ferrite + pearlite” matrix.

### 3.2. Hardness and Wear Rate

The bulk hardness values of the alloys are depicted in Table 3. The tendency in the hardness is as follows: for every group of alloys with a constant carbon concentration, the hardness increases as the boron content increases (in compliance with the total amount of hard inclusions). The near-eutectic alloys (i.e., containing 1.5 wt.% B) have a low bulk hardness (31–34.0 HRC), while the maximum bulk hardness value (53.5 HRC) is attributed to the alloy with the highest C and B concentrations (1.1C–3.5B alloy).

The experimental values of the wear rate are shown in Table 3. As seen, the *WR* values vary within the range of (2.31–5.69) × 10^−6^ g·mm^−1^·s^−1^. The best wear response was shown by two alloys with the maximum carbon content that had similar *WR* values: 1.1C–3.5B (*WR* = 2.31 × 10^−6^ g·mm^−1^·s^−1^) and 1.1C–1.5B (*WR* = 2.42 × 10^−6^ g·mm^−1^·s^−1^). The lowest wear resistance was noted for 0.3C–2.5B (*WR* = 5.69 × 10^−6^ g·mm^−1^·s^−1^). It was deduced that at any carbon content, the highest wear rate is attributed to 2.5 wt.% B. A more detailed consideration of the wear behaviour of the alloys is presented below in a subsection devoted to mathematical modelling based on the full experimental procedure.

### 3.3. Worn Surface Characteristics

The complex alloying employed in the MCCIs is intended to allow the formation of multi-component hard phases capable of effectively resisting the cutting action of abrasive particles. Figure 4a,b allow us to analyse the collision of abrasive particles with the microstructural constituents of the 0.3C–3.5B alloy, from which the trace of the abrasive particle (SiC) was not completely removed during the surface preparation. As can be seen in this figure, the very hard silicon carbide (24.5–28.2 GPa) [60] did not leave any trace on the surface of either primary inclusion or the eutectic plate, which indicates their excellent ability to withstand severe abrasion. In contrast, a deep scratch is noted in the matrix areas lying between the carboboride inclusions. In some cases, just behind the inclusion, the scratch is interrupted, vanishes, and then appears again (these sites are circled in Figure 4b). It can be presumed that the particle cannot always immediately penetrate the matrix just after sliding over the hard inclusion. The penetration resumes at some distance from the inclusion, which forms a specific “shadow zone” [61] where the matrix remains unworn because it is shielded by the carboboride inclusion. The different behaviours of carboborides and the matrix should result in uneven wear, with the predominant removal of the matrix areas leading to a specific relief on the worn surface.

Figure 4c–f show the low-magnification images, which illustrate the typical wear patterns of the worn alloys’ surfaces. Due to the preferential wear of the matrix, the carboboride inclusions are on top; they are above the level of the matrix, revealing the total picture of the structure. Herein, two main patterns of surface relief can be distinguished. The first one corresponds to near-eutectic alloys (i.e., containing 1.5 wt.% B): on their surface, the carboboride M_2_(B,C)_5_ fibres that belong to the “Chinese-script” eutectic can be clearly seen (Figure 4c,e), while the matrix around them is covered by scratches. Rough chipping and pitting on the surface are not observed in this case. The second pattern refers to hyper-eutectic alloys (i.e., with 2.5–3.5 wt.% B). It is characterised by the presence of coarse primary carboborides as protrusions towering above the worn matrix (Figure 4d,f). Multiple chipping is observed on the primary inclusions; additionally, the pits in the matrix resulting from the particles’ spalling-off can be seen. Some primary carboborides, oriented perpendicular to the surface, remained mainly free of chipping, as shown in Figure 4f. In alloy 1.1C–3.5B (Figure 4f), the coarse eutectic network (denoted as CN) is seen in some places, as it is located at an intermediate height between the matrix and the primary inclusions.

Figure 4g presents the high-magnification image of the worn surface of the near-eutectic alloy 1.1C–1.5B. It features shallow grooves in the matrix, micro-cracks, and small pits resulting from the micro-spalling of the carboboride fibres. The wear debris particles are spread over the surface as fine roundish chips of less than 0.5 μm in size. The “lips” are revealed along the grooves’ edges (see the inset in Figure 4g), as they are the preferential sites of matrix debris detachment [62]. The main characteristics of the hyper-eutectic alloys’ worn surface are the grooves of variable length and width, the “lips”, the deep pits left after carboboride spalling, and the micro-cracks on the primary inclusions (Figure 4h).

### 3.4. Regression Equation: Derivation and Analysis

The results of the abrasive wear tests were used to build model and analyse the effect of variable components (C and B) on the wear rate of the alloys. In the present study, the following type of regression equation was adopted for this purpose [54,55]:y = *a*_o_ + *a*_1×1_ + *a*_2_X_2_ + *a*_3_X_1_X_2_ + *a*_4_Z_1_ + *a*_5_Z_2_ + *a*_6_Z_1_Z_2_ + *a*_7_X_1_Z_2_ + *a*_8_X_2_Z_1_,(2)
where *a*_o_, *a*_1_, …, *a*_8_ are the coefficients; X_1_ and Z_1_ represent the coded values of the carbon content; and X_2_ and Z_2_ represent the coded values of the boron content.

To derive the regression equation, the experimental values of the wear rate (Table 3) were mathematically processed to address the standard procedure with a full factorial 3^2^ design [54,55]. The coefficients of Equation (2) were calculated using the following formulas:(3)$$a_{o}=\sum \frac{WR_{i}}{N},\;a_{1}=\sum \frac{WR_{i}X_{1i}}{N},\;a_{2}=\sum \frac{WR_{i}X_{2i}}{N},\;a_{3}=\sum \frac{WR_{i}{\left(X_{1}X_{2}\right)}_{i}}{N},\;a_{4}=\sum \frac{WRZ_{1}{}_{i}}{N},\;\;a_{5}=\sum \frac{WR_{i}Z_{2}{}_{i}}{N},a_{6}=\sum \frac{WR_{i}{\left(Z_{1}Z_{2}\right)}_{i}}{N},a_{7}=\sum \frac{WR_{i}{\left(X_{1}Z_{2}\right)}_{i}}{N},\;a_{8}=\sum \frac{WR_{i}{\left(X_{2}Z_{1}\right)}_{i}}{N},$$
where *i* = 1–9, *i* is the experiment number, and *n* = 9.

After calculating the coefficients, Equation (2) was written as follows:y = 3.48 − 0.788X_1_ + 0.292X_2_ − 0.335X_1_X_2_ + 0.035Z_1_ − 0.478Z_2_ − 0.050Z_1_Z_2_ + 0.163X_1_Z_2_ − 0.011X_2_Z_1_.(4)

With the reverse transition from Z_i_ to X_i_, the regression equation was written in its final form (*WR* × 10^−6^ g·mm^−1^·s^−1^):WR = 4.170 − 1.115X_1_ + 0.315X_2_ − 0.335X_1_X_2_ + 0.405X_1_^2^ − 1.135X_2_^2^ − 0.450X_1_^2^X_2_^2^ + 0.490X_1_X_2_^2^ − 0.035X_2_X_1_^2^.(5)

The hypothesis concerning the adequacy of Equation (5) was checked using the F-criterion. After checking, the hypothesis that Equation (5) is adequate was not rejected.

Negative values of the coefficient *a*_1_ in Equation (5) indicate a decrease in the wear rate as the carbon content increases, which was ascribed to the increase in the total amount of hard inclusions (carboborides M(C,B) and M_7_(C,B)_3_ and boron-cementite M_3_(C,B)). Positive values of the coefficient *a*_2_ indicate an increase in the wear rate under the boron effect, which is associated with the formation of coarse primary inclusions. The interactions of the variables (quadratic, third-degree, and fourth-degree) give positive or negative values of the corresponding coefficients. This reveals that carbon and boron mutually influence each other, leading to the complex non-linear character of their effect on the wear rate of the alloys.

The graphical interpretation of Equation (5) (Figure 5a) shows that it describes a non-linear response surface with “uphill” and “downhill” sections located on both sides of the extremum centerline, which corresponds to the inflection of the surface.

Figure 5b presents the projection of this surface on the “wt.% C-wt.% B” plot as a set of equidistant isolines corresponding to different *WR* values (marked next to the lines). Analysing the response surface made it possible to (a) understand the effect of C and B on the wear behaviour and (b) optimise the chemical composition of the multi-component alloys. As shown in Figure 5b, the highest *WR* value (about 5.7 × 10^−6^ g·mm^−1^·s^−1^) refers to the domain of the response surface with coordinates 0.3 wt.% C and 2.7 wt.% B. With an increase (or decrease) in the boron content relative to this value, the wear rate decreases at any carbon concentration, and thus the curve of *WR* as a function of the boron content passes through a maximum. The value of this maximum decreases as the carbon content increases. The response surface comes to its minimums at 1.1 wt.% C–3.5 wt.% B (*WR* = 2.76 × 10^−6^ g·mm^−1^·s^−1^) and at 0.86 wt.% C–1.5 wt.% B (*WR* = 2.67 × 10^−6^ g·mm^−1^·s^–1^). These combinations of carbon and boron contents (1.1C–1.5B and 1.1C–3.5B) are the most promising for the studied multi-component alloys with respect to the abrasive wear performance.

## 4. Discussion

The individual effect of the independent variables (C and B contents) was analysed using the mathematical model described in Equation (5) (Figure 6). As follows from Figure 6a, carbon continuously reduces the *WR* value at any boron content (except 1.5–1.7 wt.% B, when a slight increase in the *WR* is observed at a carbon content above ~0.85 wt.% C). When the boron content is increased to 2.5 wt.%, the curve “*WR* = *f*(wt.% C)” shifts up (to higher *WR* values) and then goes down (reflecting the decrease in the *WR* level) at higher boron concentrations. In contrast to carbon, boron affects the *WR* in a non-monotonic manner at any carbon concentration: all the curves “*WR* = *f*(wt.% B)” in Figure 6b go through the maximum corresponding to 2.5–2.7 wt.% B. With a carbon content increase, the curves shift toward lower *WR* values, while the difference in wear values decreases.

To explain the described effects of carbon and boron, the effects of the hardness and the total volume fraction of hard inclusions (M_2_(B,C)_5_, M_7_(C,B)_3_, M_3_(C,B), M(C,B)) on the wear rate should be considered first. As shown in Figure 7a, the bulk hardness (*H*) is strongly dependent on the total amount of hard compounds (*T*) according to the exponential law *H* = 19.9·exp(0.015·*T*). At the same time, there is no monotonic correlation between T and the wear rate: the curves “*WR* = *f*(*T*)” have a maximum corresponding to some *T* value specific to each carbon content (Figure 7b). A similar non-monotonic behaviour was revealed for the “bulk hardness-*WR*” correlation (Figure 7c). These observations confirm that an increase in the amount of carboborides (and the bulk hardness, accordingly) may deteriorate the wear resistance in some cases, pointing to the complexity of the wear process of the studied multi-phase alloys. In this regard, the effect of C and B on the characteristics of micro-constituents (hard inclusions and the matrix) should be considered, rather than the bulk hardness or the total amount of inclusions.

Carbon and boron are the main elements added into the alloys to form the hard phases (carbides, borides, etc.). The same behaviours are noted for the present research, as confirmed by Figure 8, which presents the effects of C and B on the total amount of hard particles. As shown in the figure, both elements increase the total volume fraction of the carboboride particles, while the effect of a particular element increases as the concentration of the other element increases. This reflects the synergy of C and B in the studied alloys, which manifests in the formation of the phases containing both elements [51].

Figure 9 and Figure 10 illustrate the individual effects of carbon and boron on the volume fractions of carboboride particles of different types. Carbon increases the amount of the Ti-rich carboboride M(C,B) (Figure 9a), as well as the Cr/Fe-rich carboborides M_7_(C,B)_3_ and M_3_(C,B)) that belong to the “rosette”, “coarse-net”, and ledeburite eutectics (Figure 9b). At the same time, carbon does not affect the number of primary inclusions of borocarbide M_2_(B,C)_5_ (Figure 9c) since it is based on the boride M_2_B_5_. Additionally, carbon slightly decreases the amount of M_2_(B,C)_5_ particles that belong to the ‘Chinese-script’ eutectic (Figure 9d). This is because carbon promotes the formation of (Cr/Fe)-rich carboboride eutectics (“rosette”, “coarse-net”, ledeburite), partially consuming the elements (Cr, W, Mo, and V) needed for M_2_(B,C)_5_ formation. This reflects the competition between boron and carbon in the formation of hard phases. Boron decreases the area fraction of the eutectic (“Chinese-script”) borocarbides M_2_(B,C)_5_ with a simultaneous increase in the fraction of the primary prismatic M_2_(B,C)_5_ inclusions (Figure 10a,b). Further, boron (at 3.5 wt.%) increases the amount of (Cr/Fe)-rich eutectic carboborides (Figure 10c), and it does not affect the amount of equiaxed Ti-rich carboborides M(C,B) (Figure 10d).

Taking into account the aforementioned results, the following explanation of the controversial effect of boron on the wear resistance of alloys can be proposed. In this research, boron is mainly responsible for borocarbide M_2_(B,C)_5_ formation. The micro-hardness of M_2_(B,C)_5_ is rather high (~2500 HV [52]), which benefits the wear resistance of the studied multi-component alloys. When the boron content is 1.5 wt.%, borocarbide M_2_(B,C)_5_ solidifies as the fine hard fibres are embedded into the softer matrix of the “Chinese-script” eutectic. Such a mutual arrangement of the matrix and hard inclusions is favourable for the wear resistance, since the dispersed fibres, being evenly distributed in the matrix, effectively protect the matrix gaps from abrasion due to the “shadow zone” effect. The closeness of the eutectic borocarbide fibres multiplies the “shadow zones”, thus reducing the depth of abrasive penetration (i.e., metal removal) (Figure 11a).

When the boron content increased to 2.5 wt.%, the studied alloys became hyper-eutectic, meaning that the solidification of coarse primary inclusions of borocarbide M_2_(B,C)_5_ occurred before the eutectic transformation. Primary particles absorbed the majority of W, Mo, and V, thus reducing the amount of eutectic (“Chinese-script”) M_2_(B,C)_5_ fibres, which became more distant from each other. The extended non-protected matrix areas outside of the “shadow zones” appeared where the abrasive particles left deep scratches, with an eventual increase in the wear rate (Figure 11a). Furthermore, the coarse primary inclusions are brittle; thus, they are prone to chipping and cracking even under specimen polishing (see the chips on the polished surface of the primary prismatoids in Figure 4g). During the wear test, their brittleness is the main detrimental factor that accelerates surface destruction. The coarse prismatic inclusions undergo intensive chipping when they collide with the abrasive particles (Figure 4b,d and Figure 11b), which sharply increases the wear rate of the alloys. Such brittle behaviour is characteristic of boride phases associated with covalent bonding in the lattice [41,42,63]. Additionally, the brittleness is aggravated due to the large size of the primary inclusions, which promotes the initiation of internal micro-cracks under solidification stresses (the cracks are shown in Figure 2b; they were caused by tensile stress when the matrix solidified inside the axial hole). The highest wear rate is attributed to the alloy 0.3C–2.5B, which contains the primary prismatoids surrounded by very sparse “Chinese-script” fibres (Figure 1b). The not-very-numerous eutectic inclusions did not appropriately protect the soft ferritic matrix, causing intensive wear with the further breakage of “denuded” primary particles (after the particle’s spalling-off, the pits appeared on the surface) (Figure 11b). Increasing the boron content to 3.5 wt.% resulted in an increase in the volume fraction of primary inclusions (by 8%), while the “Chinese-script” eutectic was replaced by the “rosette” eutectic. The latter provided a higher amount of eutectic inclusions with a closer arrangement of carboboride plates. This better shielded the matrix in the gaps between the prismatoids, leading to a decrease in the wear rate compared to the 0.3C–2.5B alloy.

The same interpretation of the effect of boron on the *WR* value is applicable for the alloys containing 0.7–1.1 wt.% C. The only distinction is that the differences in the *WR* values between 2.5 wt.% B (maximum) and 1.5 wt.% B (minimum) are much lower than those of the alloys with 0.3 wt.% C. This observation points to the importance of carbon for high-B multi-component alloys: carbon successively improves their wear response at any boron content. The most plausible reasons for this are (a) an increase in the amount of hard Ti-rich carboboride (M(C,B)) [52,64]; (b) an increase in the total amount of eutectic (Cr/Fe)-rich carboborides, which effectively protect the matrix areas between the colonies of “Chinese-script” eutectic and primary prismatoids; and (c) the alteration of the matrix structure from soft ferrite (at 0.3 wt.% C) to harder “ferrite + pearlite” or martensite (at 0.7–1.1 wt.% C). The hardness and wear resistance of the matrix are important in the context of hindering the carboboride breakage after the wear-off of the surrounding matrix.

The obtained results allow the formulation of some deductions concerning the relationship between the structure and abrasive wear resistance of the studied multi-component alloys. In general, their wear resistance is determined by the size of the hard inclusions and the uniformity of their distribution rather than by the total volume fraction of carboboride compounds. In this context, the eutectic particles are preferred because they are finer, more numerous, and located much closer to each other, which increases the extent of the “shadow zone” in the matrix. Moreover, due to their small size, the eutectic inclusions are less prone to chipping and spalling-off [61,65]. The appearance of large primary inclusions of M_2_(B,C)_5_ is accompanied by a decreasing number of “Chinese-script” eutectic particles, which deteriorates the protection of the matrix from abrasion. Most importantly, the large inclusions promote surface destruction due to their high brittleness. Carbon increases the number of eutectic inclusions through the formation of (Cr/Fe)-rich “rosette”, “coarse-net”, and ledeburite eutectics. Accordingly, in the studied amounts (0.3–1.1 wt.%), carbon improves the abrasive wear resistance in both the near-eutectic and hyper-eutectic alloys. An increase in the boron content from 1.5 wt.% to 2.5 wt.% changes the alloy’s status to hyper-eutectic, reducing the wear behaviour because of the solidification of primary inclusions at the expense of decreasing the number of eutectic inclusions.

At 3.5 wt.%, boron binds a major part of the carbide-forming elements into primary inclusions; as a result, some carbon remains unbound. This “free” carbon reacts with iron, forming the Fe-rich “coarse-net” and ledeburite eutectics with a reduced content of Cr and other alloying elements. However, even in this case, the wear resistance is lower than that of the (1.5 wt.% B)-containing alloy (see Figure 5). Only at the carbon content of 1.1 wt.%, the boron concentrations of 1.5 wt.% and 3.5 wt.% provide an approximately similar level of wear resistance, which is minimal for this study (Figure 6b). The multi-component alloy containing 1.1 wt.% C and 1.5 wt.% B is considered more promising for practical applications since the presence of large inclusions in the alloy with 3.5 wt.% B makes it extremely brittle, increasing its tendency to crack during castings and reducing its machinability. A significant increase in its wear characteristics is expected if the matrix is strengthened under heat treatment, which could be the objective of future research. In this article, the alloys were intentionally studied in the as-cast state in order to clearly reveal the effect of hard phases (borocarbides and carboborides) on their wear behaviour.

## 5. Conclusions

The microstructure and “three-body-abrasion (SiC)” wear behaviour of multi-component (wt.%) 5W–5Mo–5V–10Cr–2.5Ti-Fe(balance) alloys containing 0.3–1.1 wt.% C and 1.5–3.5 wt.% B have been investigated in the as-cast state. The following conclusions can be drawn:Depending on the boron content, the studied alloys have near-eutectic (at 1.5 wt.% B) or hyper-eutectic (at 2.5–3.5 wt.% B) structures. Near-eutectic alloys contain (in different combinations and volume ratios): (a) a (W, Mo, and V)-rich “Chinese-script” eutectic with borocarbide M_2_(B,C)_5_ fibres, (b) a (Cr/Fe)-rich “rosette” eutectic with carboboride M_7_(C,B)_3_ plates, and (c) Fe-rich “coarse-net” and ledeburite eutectics with boroncementite M_3_(C,B) plates. In hyper-eutectic alloys, coarse primary M_2_(B,C)_5_ inclusions of prismatic shape are present. All alloys comprise Ti-rich carboboride (M(C,B)) as dispersed equiaxed inclusions. As the carbon and boron contents increase, the total volume fraction of hard inclusions increases from 29 vol.% (alloy 0.3C–1.5B) to 65.7 vol.% (alloy 1.1C–3.5B). Accordingly, the bulk hardness increases from 29 HRC to 53.5 HRC.The effects of C and B on the “three-body-abrasion” behaviour were studied using the full factorial design approach (3^2^). The corresponding regression equation (with the quadratic, third-degree, and fourth-degree terms) for the wear rate was derived and analysed. It is shown that the mathematical model has a non-linear response profile, with the centreline of the maximum wear rate corresponding to 2.5–2.7 wt.% B. At this boron content, primary borocarbide M_2_(B,C)_5_ appears in the structure as coarse (tens of microns in length) brittle inclusions, while the amount of “Chinese-script” eutectic inclusions decreases.Carbon decreases the wear rate at any boron content (within their studied ranges) due to the increase in the amount of Ti-rich carboboride M(C,B) and the formation of a Cr-rich eutectic of the ”rosette” morphology (comprising carboboride M_7_(C,B)_3_), as well as “coarse-net” and ledeburite eutectics comprising boroncementite M_3_(C,B). Additionally, carbon enhances the matrix hardness, promoting its evolution from soft ferrite to harder “ferrite + pearlite” or martensite.The near-eutectic alloys were worn through the formation of shallow grooves and micro-spalling with fine debris detachment. The hyper-eutectic alloys were worn mostly through the chipping and spalling-off of the primary prismatic inclusions, accompanied by grooves of different depths. The hard phases (borocarbides and carboborides) present in the alloys effectively resisted SiC scratching. Accordingly, “shadow zones” appeared behind the inclusions where the abrasive particle could not cut the matrix plot. The dispersion and uniform distribution of hard compounds reduced the distance between them, which accordingly increased the extent of the “shadow zones”, thus decreasing the wear rate.Among the studied multi-component alloys, the lowest wear rate values corresponded to the “1.1 wt.% C–1.5 wt.% B” alloy (near-eutectic structure, 34.7 vol.% of hard inclusions) and “1.1 wt.% C–3.5 wt.% B” alloy (hyper-eutectic structure, 65.7 vol.% of hard inclusions). The first alloy is preferred for practical applications due to the absence of coarse primary inclusions, which can cause the alloy to have a high brittleness and poor machinability.


## Figures and Tables

**Figure 1 materials-16-02530-f001:**
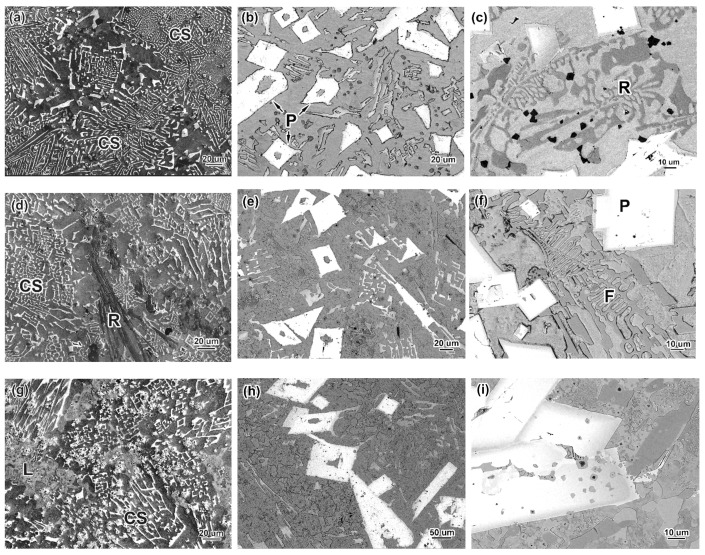
Microstructure (BSE images) of the alloys: (**a**) 0.3C–1.5B, (**b**) 0.3C–2.5B, (**c**) 0.3C–3.5B, (**d**) 0.7C–1.5B, (**e**) 0.7C–2.5B, (**f**) 0.7C–3.5B, (**g**) 1.1C–1.5B, (**h**) 1.1C–2.5B, (**i**) 1.1C–3.5B (CS is a “Chinese-script” eutectic; P is a primary carboboride M_2_(B,C)_5_; R is a “rosette” eutectic; F is a “fish-bone” eutectic; L is a ledeburite eutectic).

**Figure 2 materials-16-02530-f002:**
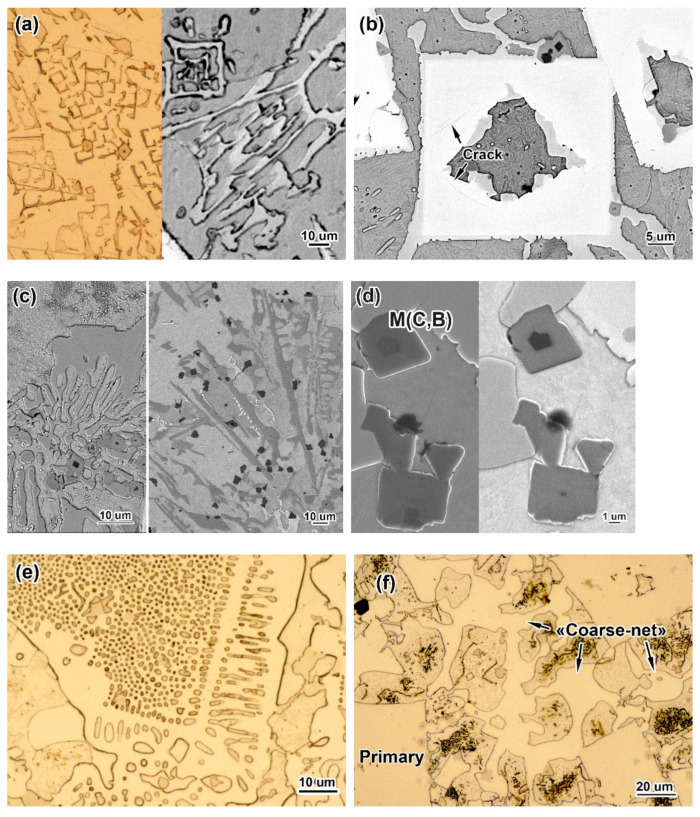
Microstructural constituents: (**a**) “Chinese-script” eutectic, (**b**) primary carboboride prismatoids M_2_(B,C)_5_, (**c**) “rosette”-like eutectic, (**d**) equiaxed angular carboborides M(C,B), (**e**) “coarse-net” eutectic, (**f**) ledeburite eutectic (**a**) (**right**), (**b**,**c**) present SEM/BSE images; (**d**) presents SEM/secondary electron (SE) (**left**) and SEM/BSE (**right**) images; (**a**) (**left**), (**e**,**f**) present OM images).

**Figure 3 materials-16-02530-f003:**
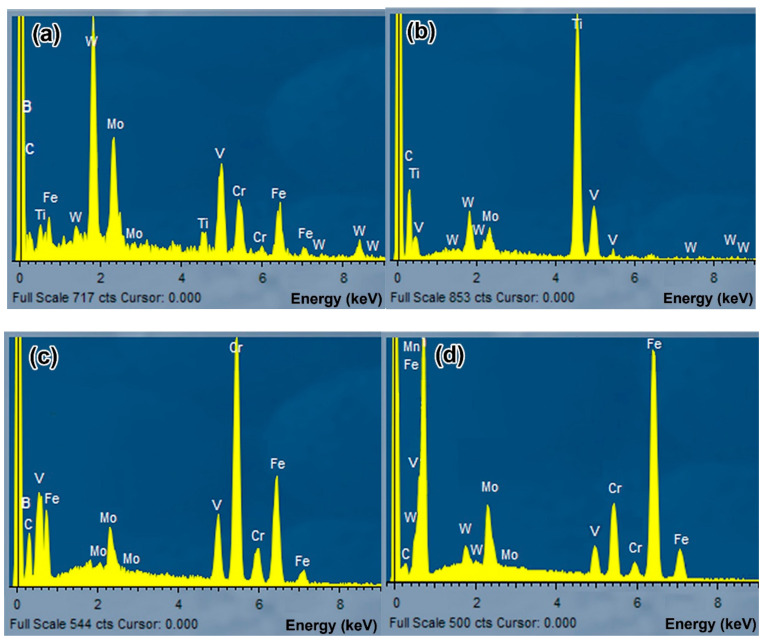
Representative EDS spectra of hard compounds: (**a**) primary M_2_(B,C)_5_, (**b**) Ti-rich equiaxed M(C,B), (**c**) (Cr/Fe)-rich eutectic M_7_(C,B)_3_, (**d**) Fe-rich eutectic M_3_(C,B).

**Figure 4 materials-16-02530-f004:**
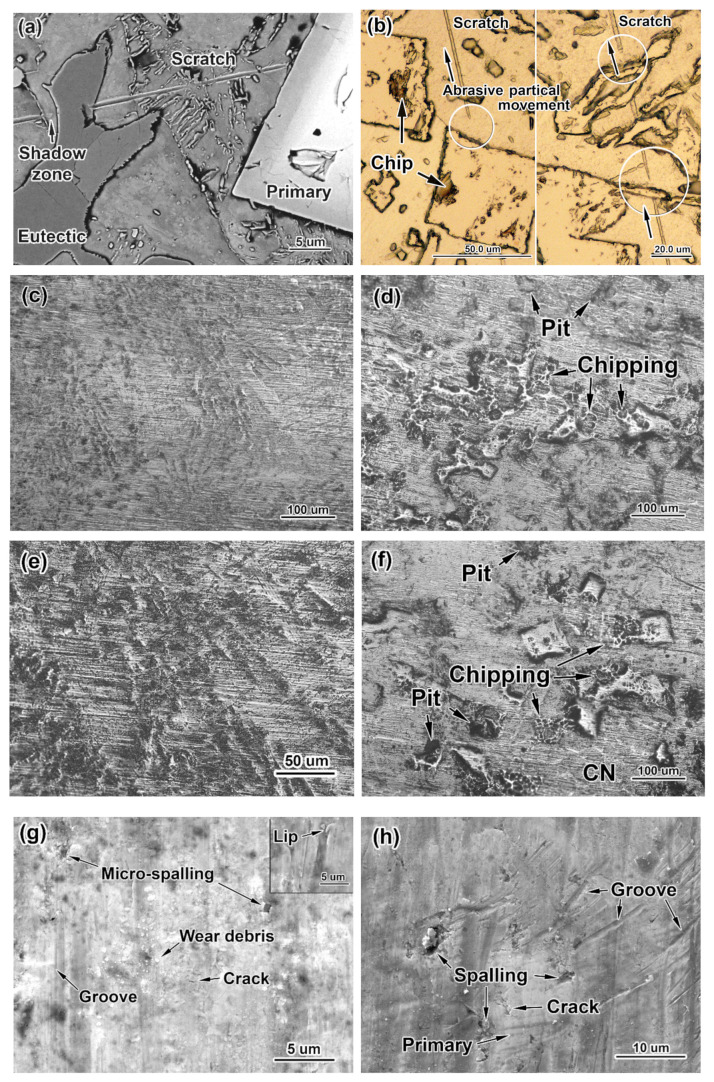
The wear behaviour of the studied alloys: (**a**) resistance of eutectic particles and primary carboboride to the abrasive sliding of SiC particles (in alloy 0.3C–3.5B); (**b**) the “shadow zones” after abrasive particle movement. The representative patterns of the worn surfaces of the alloys: (**c**) 0.3C–1.5B, (**d**,**h**) 0.3C–2.5B, (**e**,**g**) 1.1C–1.5B, (**f**) 1.1C–3.5B (**b**–**f**) are OM images; (**a**,**g**,**h**) are SEM images; CN refers to a ”coarse-network” eutectic)). Circles in (**b**) show the “shadow zones”.

**Figure 5 materials-16-02530-f005:**
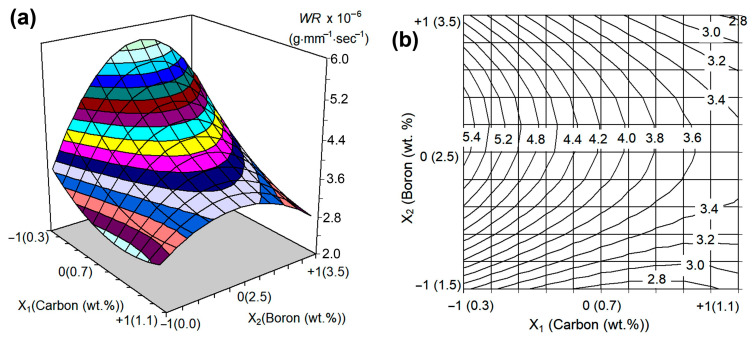
(**a**) The response surface of Equation (5) and (**b**) its projections on the “wt.% C-wt.% B” plot (the numbers next to the curve in (**b**) are the wear rate values × 10^−6^ (g·mm^−1^·s^−1^)).

**Figure 6 materials-16-02530-f006:**
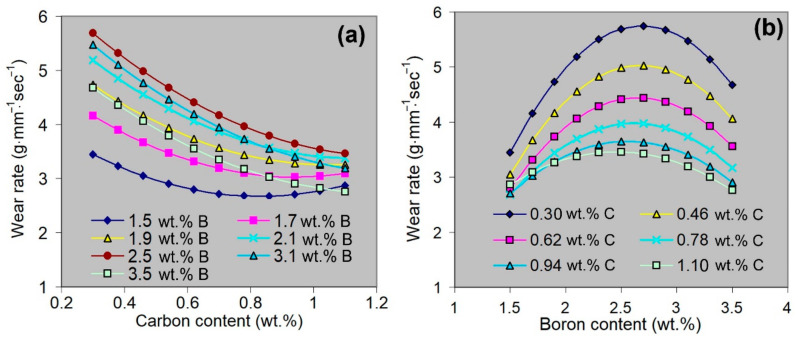
Individual effects of (**a**) carbon and (**b**) boron on the wear rate according to Equation (5).

**Figure 7 materials-16-02530-f007:**
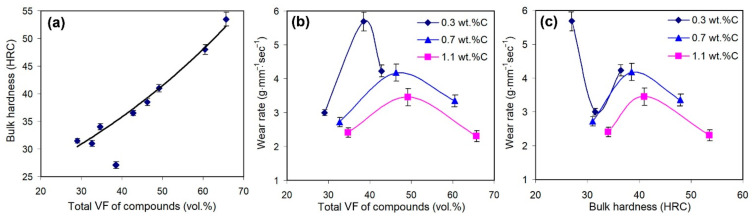
Effect of the total amount of hard compounds on (**a**) the bulk hardness and (**b**) the wear rate of the alloys. (**c**) Effect of the bulk hardness on the wear rate of the alloys.

**Figure 8 materials-16-02530-f008:**
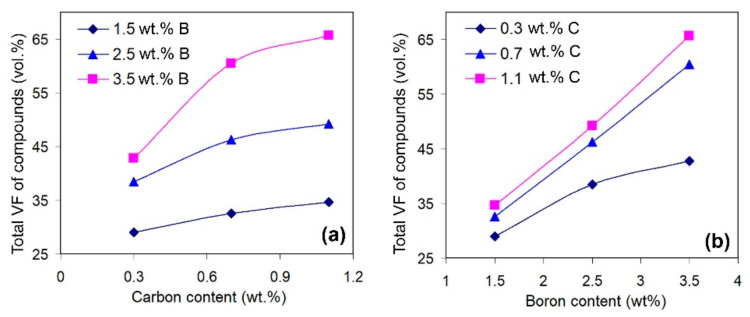
Effects of (**a**) carbon and (**b**) boron on the total amount of hard particles in the alloys.

**Figure 9 materials-16-02530-f009:**
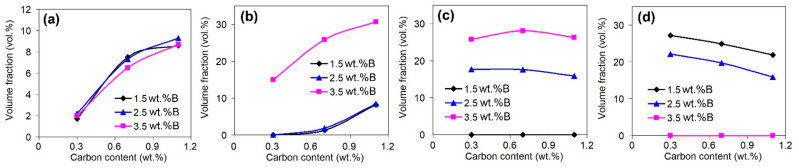
Effects of carbon on the volume fractions of carboborides: (**a**) equiaxed Ti-rich particles, (**b**) Cr-rich particles that belong to the “rosette”, “coarse-net”, and ledeburite eutectics, (**c**) primary prisms, (**d**) the particles that belong to the “Chinese-script” eutectic.

**Figure 10 materials-16-02530-f010:**
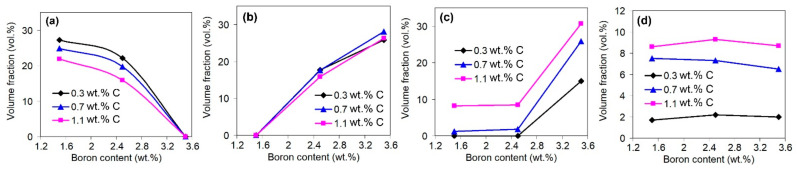
Effects of boron on the volume fractions of carboborides: (**a**) the particles that belong to the “Chinese-script” eutectic, (**b**) primary prisms, (**c**) Cr-rich particles that belong to the “rosette”, “coarse-net”, and ledeburite eutectics, (**d**) equiaxed Ti-rich particles.

**Figure 11 materials-16-02530-f011:**
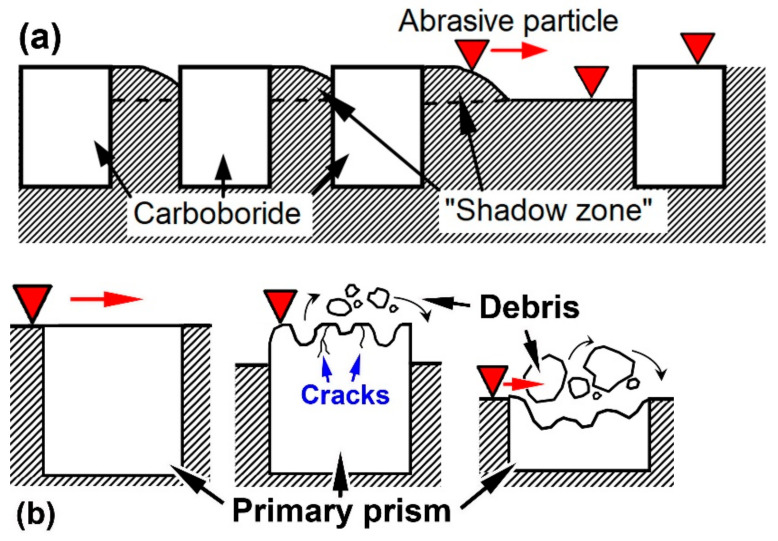
(**a**) The “shadow zones” at close-together and remote eutectic particles. (**b**) Destruction of the primary prismatic inclusion with the wear of the surrounding matrix: (**b**) the different stages of the primary borocarbide fracture (from initial state to surface chipping (small debris particles) and to volumetric spalling-off (big debris particles)).

**Table 1 materials-16-02530-t001:** Experimental domain of the selected parameters.

Variables (Factors)	Levels	Nominal Values (wt.%)	Coded Variables		Formulas of Transition from Nominal Value to X_i_ and Z_i_
			X_i_	Z_i_	
Carbon content (F_1_)	Lower	0.3	–1	1	X_1_ = (F_1_ − 0.7)/0.4 Z_1_ = 3(X_1_^2^ − 2/3)
	Middle	0.7	0	–2	
	Upper	1.1	1	1	
Boron content (F_2_)	Lower	1.5	–1	1	X_2_ = (F_2_ − 2.5) Z_2_ = 3(X_2_^2^ − 2/3)
	Middle	2.5	0	–2	
	Upper	3.5	1	1	

**Table 2 materials-16-02530-t002:** The matrix of the experiment and the chemical compositions of alloys (in all alloys, Fe is balance).

Alloy Designation	Matrix		Content (wt.%)									
	F_1_(C)	F_2_(B)	C	B	Si	Mn	Cr	Mo	V	W	Ti	Al
0.3C–1.5B	–1	–1	0.22	1.68	0.95	1.06	10.39	4.80	5.21	5.57	2.55	0.15
0.3C–2.5B	–1	0	0.25	2.70	1.05	1.21	9.85	5.14	5.01	4.68	2.83	0.05
0.3C–3.5B	–1	1	0.30	3.62	1.14	0.88	10.32	5.19	5.35	5.35	2.78	0.08
0.7C–1.5B	0	–1	0.77	1.62	1.12	1.16	10.45	5.38	4.97	5.84	2.93	0.05
0.7C–2.5B	0	0	0.72	2.75	1.10	0.90	10.35	5.57	5.78	5.05	2.60	0.04
0.7C–3.5B	0	1	0.70	3.61	1.18	1.07	10.21	4.63	5.40	4.67	2.71	0.08
1.1C–1.5B	1	–1	1.20	1.59	1.07	1.10	10.41	4.48	5.37	5.42	2.38	0.10
1.1C–2.5B	1	0	1.11	2.73	1.10	1.07	10.36	4.69	5.26	4.85	2.43	0.14
1.1C–3.5B	1	1	1.13	3.57	1.06	1.03	9.94	4.08	4.79	4.50	2.39	0.11

**Table 3 materials-16-02530-t003:** The structure characteristics, bulk hardness, and wear rate values of the alloys. CS refers to a “Chinese-script” eutectic; PP refers to a primary prismatoid M_2_(B,C)_5_; R refers to a “rosette” eutectic; EC refers to an equiaxed cuboid particle M(C,B); L refers to a ledeburite eutectic; and CN refers to a “coarse-net” eutectic. The structure of the matrix is given in brackets (F, P, and M refer to ferrite, pearlite, and martensite, respectively).

Alloys	Structure Characteristics	Volume Fractions (Vol.%)		Bulk Hardness (HRC)	Wear Rate (×10^−6^ g·mm^−1^·s^−1^)
		Structural Constituents	Hard Particles		
0.3C–1.5B	Eutectic	CS (98.3), EC (1.7) (F)	29.0	31.5 ± 0.4	3.00 ± 0.10
0.3C–2.5B	Hyper-eutectic	CS (80.1), PP (17.7), EC (2.2) (F)	38.5	27.0 ± 0.3	5.69 ± 0.51
0.3C–3.5B	Hyper-eutectic	R (72.3), PP (25.8), EC (2.0) (F)	42.8	36.5 ± 0.3	4.23 ± 0.19
0.7C–1.5B	Eutectic	CS (89.8), R (2.7), EC (7.5) (F)	32.6	31.0 ± 0.4	2.72 ± 0.11
0.7C–2.5B	Hyper-eutectic	CS (71.2), PP (17.6), R (3.9), EC (7.3) (M)	46.3	38.5 ± 0.5	4.17 ± 0.37
0.7C–3.5B	Hyper-eutectic	R (65.4), PP (28.1), EC (6.5) (F + P)	60.5	48.0 ± 0.8	3.35 ± 0.09
1.1C–1.5B	Eutectic	CS (79.0), L (10.3), R (2.1), EC (8.6) (F)	34.7	34.0 ± 0.5	2.42 ± 0.15
1.1C–2.5B	Hyper-eutectic	CS (57.5), PP (15.9), CN (17.3), EC(9.3) (M)	49.2	41.0 ± 0.6	3.46 ± 0.27
1.1C–3.5B	Hyper-eutectic	CN (65.0), PP (26.3), EC (8.7) (F + P)	65.7	53.5 ± 1.1	2.31 ± 0.17

## Data Availability

Not applicable.

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
