# Peer review of "Investigations of Abrasive Wear Behaviour of Hybrid High-Boron Multi-Component Alloys: Effect of Boron and Carbon Contents by the Factorial Design Method"

_materials, 2023, doi:10.3390/ma16062530_

Round 1

Reviewer 1 Report

1. There are too many keywords, normally, five keywords are enough.

2. Longitudinal coordinates should be added in Fig.3.

3. What’s the meaning of (W,Mo,V)-rich borocarbide M2(B,C)5? For clarify the kind of inclusions (such as borocarbide, borocementite, or the others), other analysis such as XRD, EBSD or TEM observation should be added.

4. As for the micro-hardness, how many positions were tested? Error bars should be added in Fig.7.

5. Some minor errors should be corrected, such as 2.31…5.69.

Author Response

Comment 1: There are too many keywords, normally, five keywords are enough.

Response: The number of keywords is reduced to five.

Comment 2. Longitudinal coordinates should be added in Fig.3.

Response: Figure 3 shows the EDX spectra. Along y-axis the EDX counts value should be plotted. However, the software of the EDX detector (INCA) does not illustrate the y-axis with values on the screen. Instead, the maximal count number is displayed at the bottom part of the diagram, along the x-axis.

To address this comment, we modified Figs. 3a-3d by adding the maximal count number to the plots. Previously they were removed in order to reduce the size of the figures.

Comment 3. What’s the meaning of (W,Mo,V)-rich borocarbide M2(B,C)5? For clarify the kind of inclusions (such as borocarbide, borocementite, or the others), other analysis such as XRD, EBSD or TEM observation should be added.

Response: The present article is a continuation of our previous works (see references [51-53]), in which the features of solidification, phase-structural state, and phase elemental distribution in the studied alloys are investigated in detail. Those articles contain the results of the XRD and EDX characterization of different phases, including carboboride M2(B,C)5. In order to avoid repeating this vast array of data in the present article, we refer in the text to previously published works.

The approximate contents of alloying elements (W, Mo, V, Cr, Ti) in various carboboride phases are indicated within subsection 3.1 and confirmed by the EDX-spectra presented in Figure 3.

Comment 4. As for the micro-hardness, how many positions were tested? Error bars should be added in Fig.7.

Response: Actually, micro-hardness was not measured in the present work therefore mentioning of micro-hardness testing is removed from the Materials section. In present work, we measured the bulk hardness according to the Rockwell method. Five measurements were performed on each of the three specimens intended for wear test (a total of 15 measurements for each alloy). The results were then averaged for all the specimens. This information is added to the text (please see page 4, the 1st paragraph).

Error bars are added to Figs.7a, 7b, and 7c.

Comment 5. Some minor errors should be corrected, such as 2.31…5.69.

Response: The text was subjected to the extensive professional proof-reading (the Certificate is submitted with the revised paper); all the grammar errors and typos are corrected.

Reviewer 2 Report

-Generally, English grammar is very poor. There is a miscommunication in many places. It should be reviewed again.

-I think there is a problem with the scaling of Figure 1 and 2. Visuals with 10 um and 20 um should be checked and the points that we need to focus on in the figures should be specified.

-In the EDS analysis shown in Figure.3, only the relevant components should be shown instead of all components. In addition, the data obtained should be explained with the support of the literature.

-Figure.4 c-h images are very bad, something is not understood. It can be updated again if possible.

- It should be stated where the regression equations expressed in Line 348 were taken from or adapted. It also looks very complex as it is, it can be simplified.

-The figure shown in a in Figure 5 is sufficient, why did you put b? The evaluations to be made between the data lines are insufficient. Which interval is sufficient or which interval is insufficient?

-Most of the issues expressed between line 463-551 can be shifted to the introduction or methods section. It's pointless to be here.

Author Response

Comment 1: Generally, English grammar is very poor. There is a miscommunication in many places. It should be reviewed again.

Response: The text was subjected to the extensive professional proof-reading (the Certificate is submitted with the revised paper); all the grammar errors and typos are corrected.

Comment 2: In the EDS analysis shown in Figure.3, only the relevant components should be shown instead of all components. In addition, the data obtained should be explained with the support of the literature.

Response: Figures 3a-3d are modified accordingly.

Comment 3: I think there is a problem with the scaling of Figure 1 and 2. Visuals with 10 um and 20 um should be checked and the points that we need to focus on in the figures should be specified.

Response: Figures 1 and 2 were checked and the scale bar of Fig. 1f was corrected. The structural components (“Chinese-script” eutectic, primary carboboride M2(B,C)5, “Rosette” eutectic, Ledeburite eutectic) were marked in Figs. 1a, 1b, 1c, 1d, 1f and 1g.

Comment 4: Figure.4 c-h images are very bad, something is not understood. It can be updated again if possible.

Response: Figures 4c-h give the total view of the worn surfaces of alloys. They were made using an optical microscope by purpose, in order to reveal surface topography and wear mechanisms due to the optical effects and shadows. Specifically, they show that carboborides inclusions rise over a more worn matrix. These figures allow clearly identify the fracture and chipping of primary inclusions (in hypereutectic alloys), or the absence of such fracture in alloys with a near-eutectic structure. We kindly propose to leave these images as the key illustration reflecting the nature of the abrasive wear of the studied alloys.

Comment 5: It should be stated where the regression equations expressed in Line 348 were taken from or adapted. It also looks very complex as it is, it can be simplified.

Response: The approach of the full design experiment and the regression equation (2) were exactly adopted from: Novik, F.S., Arsov, J.B. Optimization of metal technology processes by methods of planning experiments, Mashinostroenie, 1980 (in Russian). This new reference is added to the Reference List.

Equation (2) is intended to describe complex response surfaces with variable relief, therefore it contains terms of the quadratic, the third, and the fourth degrees. It does not make sense to simplify it since in this case the procedure for processing the results will be violated and the resulting model will contain a big error.

Comment 6: The figure shown in a in Figure 5 is sufficient, why did you put b? The evaluations to be made between the data lines are insufficient. Which interval is sufficient or which interval is insufficient?

Response: Although Figures 5a and 5b are derived from the same equation (2), they bring different information. Figure 5a illustrates the general relief of the response surface, while Figure 5b gives the specific distribution of WR values on the concentration plane. Therefore, we propose to leave both drawings.

In order to increase the information content of Figure 5b, we reduced the interval in values between the isolines from 0.3 to 0.2. This allows to more accurately determining the areas of the diagram that correspond to the minimum and maximum wear rates.

Comment 7: Most of the issues expressed between line 463-551 can be shifted to the introduction or methods section. It's pointless to be here.

Response: The information presented on lines 463-551 is a discussion of the experimental results obtained in terms of the effect of boron and carbon content on the abrasive wear resistance of alloys. In particular, the change in the microstructure as the boron content increases is analyzed focusing on the appearance of large primary inclusions of complex carboboride M2(B,C)5, which affects the wear mechanisms due to their high brittleness. In support of this reasoning, the possible schemes of wear mechanisms are suggested depending on the microstructure features. Summarizing, a conclusion was made about the optimal contents of C and B in terms of wear resistance and mechanical properties. We suppose that all these considerations can not be moved to the Method section or to the Introduction since they are the main outcomes of the present works.

Round 2

Reviewer 1 Report

1.Page 4, Lin 148: Wear testing should be hardness testing.

2. How to confirm (W,Mo,V)-rich borocarbide M2(B,C)5? For clarify the kind of inclusions (such as borocarbide, borocementite, or the others), other analysis such as XRD, EBSD or TEM observation should be added.

3. Longitudinal coordinates did not be added in Fig.3.

Author Response

  1. Comment: Page 4, Lin 148: Wear testing should be hardness testing.

 Response: Actually, there was no mistake in this place. The meaning of this sentence was that the hardness measurements were carried out exactly on the specimens which were intended (used) for wear testing. By this sentence, we pointed out that the hardness value and wear rate were obtained from the same specimens. This allowed for strengthening the reliability of the “hardness-wear rate” correlation.

However to avoid any misunderstanding, we modified the sentence as follows: “Five hardness measurements were performed on each of three different specimens (a total of 15 measurements for each alloy), with further averaging of the results.” (Please see page 4).

  1. Comment: How to confirm (W,Mo,V)-rich borocarbide M2(B,C)5? For clarify the kind of inclusions (such as borocarbide, borocementite, or the others), other analysis such as XRD, EBSD or TEM observation should be added.

Response: We agree with the Reviewer that types of borocarbide inclusions in the studied alloys should be justified. However, the phase identification was fulfilled and confirmed in our previously published works [51-53], presented in the Reference list:

  1. Efremenko, V.G.; Chabak, Yu.G.; Shimizu, K.; Golinskyi, M.A.; Lekatou, A.G.; Petryshynets, I.; Efremenko, B.V.; Halfa, H.; Kusumoto, K.; Zurnadzhy, V.I. The novel hybrid concept on designing advanced multi-component cast irons: Effect of boron and titanium (Thermodynamic modelling, microstructure and mechanical property evaluation). Mater. Charact. 2023, 197, 112691.
  2. Chabak, Yu.G.; Shimizu, K.; Efremenko, V.G.; Golinskyi, M.A.; Kusumoto, K.; Zurnadzhy, V.I.; Efremenko, A.V. Microstructure and phase elemental distribution in high-boron multi-component cast irons. International Journal of Minerals, Metallurgy, and Materials 2022, 29, 78-87.
  3. Chabak, Yu.G.; Golinskyi,М.А.; Efremenko, V.G.; Shimizu,К.; Halfa, H.; Zurnadzhy, V.І.; Efremenko, B.V.; Kovbasiuk, Т.М. Phase constituents modeling in hybrid multi-component high-boron alloy. Physics and Chemistry of Solid State 2022, 23, 714-719.

 These papers are focused on the characterization of the structure of the same alloys studied; they include a detailed analysis of the “Thermo-Calc”, XRD and SEM/EDS results proving the type of different hard inclusions found in the alloys. We repeatedly refer to these articles in the text, for example:

- page 3: “The structure characterisation and thermodynamic analysis of “hybrid” MCCIs with variable carbon (0.3–1.1 wt.%) and boron (1.5–3.5 wt.%) contents are described in detail in [51-53];

- page 4: “The data from the XRD/EDS study of the phase constituents of the alloys were adopted from [51, 52]”;

- page 5: “The fibres were characterised in earlier works [51, 52] as borocarbide M2(B,C)5 ….”;

- page 6: “The colonies of the ‘”Rosette” eutectic are the plates (carboboride M7(C,B)[51])….”, etc.

Furthermore, in the manuscript we present the phase chemical compositions of different phases, as follows:

- page 5: “…borocarbide M2(B,C)containing W (12-30 wt.%), Mo (14-19 wt.%), V (15-18 wt.%), Cr (10-21 wt.%), and Ti (1-5 wt.%) (the exact  content of each element depends on C and B concentrations in alloy);

- page 6: “The lamellae of the ‘”Rosette”-like eutectic differ from the primary inclusions due to their darker BSE contrast, indicating their enrichment in chromium (~31–45 wt.%) and iron (~30–40 wt.%) [51, 52].”;

- page 7: “Ti-based carboboride (M(C,B)) particles that were the first to crystallise from the melt due to the addition of 2.5 wt.% Ti. The titanium content varies in M(C,B) in a rather wide range (54–72 wt.% [51, 52])”;

- page 7: “a coarse network with the lowered chromium content (20-22 wt.%) [51] in the eutectic plates.”

- page 7: “The lamellae of the ”Coarse-net” eutectic are heterogeneous in terms of their Cr and Fe contents, which vary from 12–22 wt.% and 67–75 wt.%, respectively [51].”

 Therefore, we believe that there is no need to repeat in this paper the previously published data on the identification of hard phases (including M2(B,C)5)). This could cause an unreasonable increase in the length of the article which is focused on wear resistance rather than on phase characterization.

We do hope that our arguments will be convincing for the Reviewer.

  1. Comment: Longitudinal coordinates did not be added in Fig.3.

Response: Figure 3 is corrected, the longitudal coordinate designation (Energy (keV)) is added.

Reviewer 2 Report

Thanks to the authors for their corrections and replies. The article can be accepted as it is.

Author Response

Thank a lot for your review.